# A Potent Antibacterial Peptide (P6) from the De Novo Transcriptome of the Microalga *Aureococcus anophagefferens*

**DOI:** 10.3390/ijms252413736

**Published:** 2024-12-23

**Authors:** Kexin Zhang, Xiaoting Yin, Yu Huang, Chao Liu, Qingchun Zhang, Qing Liu, Senyu Wang, Wenwu Fei, Qiong Shi, Limei Qiu

**Affiliations:** 1School of Marine Biology and Engineering, Qingdao Agricultural University, Qingdao 266109, China; zhangkexin9822@163.com (K.Z.); wangsenyu@qdio.ac.cn (S.W.); 15698293839@163.com (W.F.); 2Key Laboratory of Breeding Biotechnology and Sustainable Aquaculture (CAS), Institute of Oceanology, Chinese Academy of Sciences, Qingdao 266000, China; xiaoting.yxt@foxmail.com (X.Y.); liuqing@qdio.ac.cn (Q.L.); 3Shandong Province Key Laboratory of Experimental Marine Biology, Institute of Oceanology, Chinese Academy of Sciences, Qingdao 266000, China; 4Key Laboratory of Breeding Biotechnology and Sustainable Aquaculture, Institute of Hydrobiology, Chinese Academy of Sciences, Qingdao 430072, China; huangyu@ihb.ac.cn; 5Key Laboratory of Marine Ecology and Environmental Sciences, Institute of Oceanology, Chinese Academy of Sciences, Qingdao 266000, China; liuchao@qdio.ac.cn (C.L.); qczhang@qdio.ac.cn (Q.Z.); 6Laboratory of Aquatic Genomics, College of Life Sciences and Oceanography, Shenzhen University, Shenzhen 518057, China

**Keywords:** *Aureococcus anophagefferens*, antimicrobial peptides (AMPs), high-throughput prediction, antimicrobial activity, de novo transcriptome

## Abstract

Marine microalgae are a rich source of natural products, and their amino acid-based antimicrobial agents are usually obtained by enzymatic hydrolysis, which is inefficient and limits the research on antimicrobial peptides (AMPs) from microalgae. In this study, *Aureococcus anophagefferens* is used as a model to predict antimicrobial peptides through high-throughput methods, and 471 putative peptides are identified based on the de novo transcriptome technique. Among them, three short peptides, P1, P6, and P7 were found to have antimicrobial activity against *Escherichia coli*, *Staphylococcus aureus*, *Micro1coccus luteus*, and yeast *Pichia pastoris*, and they showed no hemolytic activity even at higher concentrations up to 10 mg/mL. Especially P6, a 12-amino acid peptide with three positive charges, which exhibited the most significant microbicidal effect with the lowest MIC of 31.25 μg/mL against *E. coli*, and electron microscope observations showed the surface of P6 treated *E. coli* with granular protrusions and ruptures, suggesting that it likely caused cell death by directly destroying the bacterial cell membrane. This study may enrich the database of microalgal AMPs and demonstrate an efficient process for searching and validating microalgal source AMPs by combining computer analysis with bioactivity experiments.

## 1. Introduction

Studies have shown that deaths associated with 33 common bacterial pathogens accounted for 13.6% of all deaths worldwide in 2019 [1]. Although the discovery and use of antibiotics have been very effective in the prevention and treatment of various infectious diseases, revolutionizing human medicine, agriculture, and food production, the overuse of antibiotics has promoted the development of drug-resistant bacteria and made antibiotics themselves less effective [2]. At this time, in order to fight infectious diseases, the need to find the most effective natural antibacterial substances is becoming more and more urgent. Natural substances have a variety of biological activities and are effective therapeutic agents for the treatment of a variety of diseases. In particular, some herbs have been widely used as antibacterial agents since prehistoric times because of their low cost and easy availability [3]. Antimicrobial peptides (AMPs) are a special group of natural products that protect the host from infections and are the most promising alternatives to antibiotics. A considerable amount of work has been performed to isolate natural AMPs from various organisms that by the end of December 2022, the AMP database APD3 (https://aps.unmc.edu/) contained 3569 AMPs and proteins, including 3067 natural AMPs with activity, of which more than two-thirds of AMPs originate from animals and bacteria, with only a small portion coming from plants [4], that is, many natural sources remain underexplored, which might partly explain the current scarcity of natural antimicrobial compounds.

The ocean has plentiful biological resources, making it a good source for novel active substances of interest. Marine organisms and their metabolites are the focus of worldwide efforts for the discovery of novel biologically active products, which are the foundation of the future bioeconomy and particularly as promising drugs. In this context, marine-derived AMPs have sparked heightened interest due to their remarkable characteristics, namely their outstanding stability, excellent solubility, and potent bactericidal effects [5]. However, current research trends primarily concentrate on extracting AMPs from crustacean and mollusk sources, while AMPs derived from plant-like marine organisms, such as algae and microalgae, remain largely unexplored. Microalgae, a highly diverse phylum of photosynthetic life forms inhabiting aquatic ecosystems across the globe, possess an extraordinary aptitude for synthesizing a myriad of bioactive compounds. Their distinct metabolic pathways and adaptive mechanisms afford them a unique potential to generate a wide array of antibacterial substances. For example, SP-1 from the microalga *Spirulina platensis* has shown potent activity against various bacteria and fungi [6]. Nostocyclin, derived from the cyanobacterium *Nostoc* sp., exhibits a different mode of antibacterial action compared to conventional antibiotics [7]. More importantly, microalgae-derived AMPs have offered numerous application prospects. For example, microalgae AMPs can be utilized as natural alternatives to synthetic pesticides or additives, providing effective and sustainable solutions for plant disease management and food preservation [8,9]. Also because of the strong bactericidal, antioxidant, surfactant, and other activities of microalgae AMPs, it is utilized in cosmetics, medicine, and other industries [10]. Additionally, they are utilized in wound healing due to their antimicrobial properties and ability to aid in tissue regeneration [11]. They are also employed in drug delivery systems, anti-biofilm coatings, and the development of antimicrobial materials for medical devices [10,11]. In a word, the unique properties of AMPs derived from microalgae make them attractive candidates for developing sustainable and natural alternatives in various fields, contributing to global health, agriculture, and food safety.

Despite the considerable application potential of the microalgae derived AMPs, it is noted that only a limited number of AMPs from microalgae have been identified and characterized to date. Most of them are often obtained through enzymatic hydrolysis techniques, which involve breaking down complex proteins within microalgal biomass into smaller bioactive peptides. As an example, the whole protein of *Saccharina longicruris* was hydrolyzed by using enzymes to obtain proteolysates by controlling temperature, pH, and the ratio of enzymes to substrates [12]. Ultrafiltration was then carried out on the hydrolysates produced by enzymatic hydrolysis, and the inhibitory activity of hydrolysate components greater than 10 kDa against *Staphylococcus aureus* was verified in vitro, but this method resulted in a complex mixture of compounds, making it difficult to isolate and identify specific active antimicrobial components [12]. In addition, although the hydrolysis conditions are mild and easy to control, the selection of different enzymes, the degree of hydrolysis, and the concentration of substrates will affect the polypeptides obtained. Therefore, the traditional enzymatic hydrolysis method is not an effective way to find new AMPs in microalgae, and the paucity of known microalgal AMPs underscores the untapped resource that these organisms represent in the search for novel antimicrobial agents.

In recent years, with the continuous development of bioinformatics technology, the high-throughput prediction of AMPs from genomic and transcriptomic datasets using bioinformatics tools and algorithms has emerged as one of the practical and effective methods to discover novel AMP sequences [13,14]. Whole-genome sequencing can sometimes be challenging due to technical and financial constraints, whereas the transcriptome sequencing of non-model organisms is relatively straightforward and rapid. This approach allows researchers to analyze non-model organisms through de novo transcriptome assembly, which greatly expands the scope for discovering AMPs [15]. Currently, a number of new AMPs have been identified through high-throughput prediction using datasets from various organisms, including sturgeon [16], ciona [17,18], anemone [19], and *Leymus arenarius* [20]. As an example, 37 candidate AMPs were discovered in sturgeon, and 11 of them were found to have inhibitory effects on *Escherichia coli* after activity verification [16]. These findings highlight the potential of high-throughput and accurate screening strategies in accelerating the discovery of AMPs across different organisms. However, despite these advancements, the application of high-throughput prediction in microalgae remains largely unexplored. The ability of microalgae to synthesize a wide range of bioactive compounds and their ecological importance in aquatic ecosystems suggest that they could be valuable targets for AMP discovery. Nevertheless, the practicability of the process and the efficiency of AMP discovery in microalgae species remain to be fully understood. Further research is needed to optimize bioinformatics tools and algorithms specifically for microalgae, which will potentially unlock new avenues in the development of natural antimicrobial agents.

Pelagophyte *Aureococcus anophagefferens* is a cosmopolitan picoplanktonic species, having a nearly spherical shape diameter of approximately 2–3 µm. It is known to form intense harmful algal blooms (HABs) with high cell abundance exceeding 10^9^ cells/L in coastal waters [21]. The HAB caused by *A. anophagefferens* is called brown tide because the high abundance of proliferated algae causes the sea water to turn brown and adversely affect the feeding behavior of bivalves [22]. The brown tides of *A. anophagefferens* have been documented in the eastern coastal waters of the United States, a bay in South Africa, and the waters of the Bohai and Yellow Seas, China [23,24,25,26]. Accumulated evidence suggests that *A. anophagefferens* prefers environments with high levels of dissolved organic matter (DOM). The presence of high DOM content in inshore regions is particularly favorable for its growth [5]. Compared to competing phytoplankton, *A. anophagefferens* has unique genes for light collection, organic matter utilization, and selenium-requiring enzymes relative [27]. For instance, under nitrogen restriction, ammonia transporters, acetamidase or formamidase, and the expression of peptidases are increased [28]. These genetic adaptations likely assist *A. anophagefferens* in adapting to adverse environments and enhance its competitive advantage over other phytoplankton species. Moreover, the ability to produce unique and effective molecules indicates the high possibility of the presence of natural alternatives to synthetic antibiotics and other antimicrobial agents. Further research into the genetic capabilities of *A. anophagefferens* is therefore supposed to uncover novel AMPs with broad-spectrum antimicrobial activities.

As the brown tides in the coastal waters of the United States and China are suggested to be caused by the respective native strains [29], there are differences in ecophysiology, toxicity, and genetics among different geographical strains of *A. anophagefferens* [22,30]. In this study, a strain of *A. anophagefferens* is isolated from the coastal waters of Qinhuangdao and cultured under various environmental stresses to maximize the activation of different gene expressions. Its de novo transcriptome is assembled, and high-throughput prediction is conducted to create a database of putative AMP fragments. The selective short peptides are synthesized to verify their antimicrobial activity, and the mechanism of their antimicrobial effects is analyzed based on their predicted structure and electron microscopic observations.

## 2. Results

### 2.1. Sequencing and De Novo Assembly of A. anophagefferens Transcriptome

Firstly, five transcriptome fragment libraries (details in Section 4.1) were built with microalgae mRNA and were sequenced to generate a total of 206,662,594 raw reads. The read numbers of each library are shown in Table 1. Then, adapter trimming and low-quality sequences were removed, and the resultant total clean reads was 202,543,652 (98%) and 202 M clean reads of 150 bp in length. High Illumina quality control scores of Q30 and Q20 were observed in all sequenced samples, which is indicative of high-quality data (Table 1). The sequencing data merging into around 30 G clean data were assembled with Trinity, a software package for transcriptome assembly. The obtained contigs were clustered into 47,212 unigenes, of which 39,420 contained a typical ORF and could be translated into protein sequences.

### 2.2. Characteristic Analysis of Candidate AMPs

After blasting against the APD3 database and removing alignments of less than 80% of alignment lengths, a total of 1687 sequences were selected for further AMP prediction using different models and software. The 471 overlaps among different methods were considered potential AMPs for the downstream analysis. Because there were no typical full-length AMPs identified, we mainly focused on the peptides that probably lysed from the N- or C- terminus of protein, and 52 peptides were screened out (Appendix A). As a result, most of them were found to be lysed from the N-terminus, among which the five shortest polypeptides with a net charge greater than three were chosen for further analysis (P1–P5). Moreover, P6 and P7 were also selected, as they were the only two peptides broken off from the C- terminus. The physicochemical traits of the seven peptides, such as the isoelectric point, net charge, and grand average of hydropathicity, were examined (Table 2). Although they were all positively charged (3~5) and had high isoelectric points (10.62~12.70), two peptides (P3 and P4) had a very low hydrophobic moment (0.36 and 0.19, respectively), which may affect the penetrating of the cell membrane [8]. Amphiphilicity was also essential for the correct folding of the AMPs [9], and the calculated amphiphilicity indexes of peptide P6 and P7 were highest and P2 was lowest.

### 2.3. Antimicrobial Confirmation

Growth curves of three bacterial strains, namely *E. coli*, *S. aureus*, and *Micrococcus luteus*, in the absence or presence of peptides were obtained through optical density (OD600) measurements. Preliminary experiments indicated that P2-P5 exhibited no or negligible antibacterial activity (Supplementary Appendix A), while P1, P6, and P7 demonstrated significant inhibitory activity against these bacteria, prompting further testing using a twofold serial dilution method. As shown in Figure 1, there was no bacterial growth in the phosphate-buffered saline (PBS) and peptide groups, indicating that the experimental system was uncontaminated and that bacterial growth occurred under normal conditions. The results displayed that P1 could not inhibit the growth of *M. luteus* but inhibited the growth of *E. coli* and *S. aureus*, while P7 had similar inhibitory abilities against *E. coli* but showed only very weak inhibitory effects on *S. aureus* and *M. luteus* in their early stages of growth. Both P1 and P7 did not completely inhibit the three bacteria within the tested concentration range (0–500 μg/mL); in contrast, P6 exhibited the strongest inhibitory effects on *E. coli*, *S. aureus*, and *M. luteus* at concentrations as low as 7.8125 μg/mL, achieving complete inhibition with minimum inhibitory concentrations (MICs) of 31.25 μg/mL, 125 μg/mL, and 125 μg/mL, respectively (Table 3; Figure 1).

In addition to testing against bacteria, the bioactivity of the seven synthesized peptides against *Pichia pastoris* was also assessed. Consistent with the bacterial results, P2–P5 showed no inhibitory effect (Appendix A), whereas only P1, P6, and P7 significantly exhibited antifungal activity (Figure 2; Table 3). Among them, P6 demonstrated the most significant inhibitory activity against *P. pastoris*, with an MIC of 62.5 μg/mL, which was notably lower than the concentrations required for P1 (250 μg/mL) and P7 (500 μg/mL) to achieve complete yeast inhibition. Clearly, P6 was a robust antimicrobial peptide characterized by its broad-spectrum activity, probably inhibiting fungal, Gram-positive, and Gram-negative bacterial microorganisms effectively.

### 2.4. Hemolytic Activity

The hemolytic activities of P1, P6, and P7 were evaluated to support their potential future applications. In our preliminary tests, dimethyl sulfoxide (DMSO) at various concentrations induced different levels of cell membrane disruption and cell lysis, demonstrating a concentration-dependent effect (Figure 3A); meanwhile, P6 and P7 (0.3125 to 2.5 μg/mL) did not exhibit any hemolytic activity (Appendix A). Even at higher concentrations of 10 mg/mL and 5 mg/mL, P1, P6, and P7 displayed no hemolytic effect, similar to the negative control (0.9% NaCl), which had a hemolysis rate below 2% (Figure 3B). In contrast, 50% DMSO served as a positive control and exhibited a hemolysis rate of approximately 50%. These results demonstrated that P1, P6, and P7, despite their bactericidal effects at high concentrations, were safe for blood cells and exhibited no cytotoxicity.

### 2.5. Similarity and Structural Analysis of the Three Bactericidal AMPs

The alignment of P1, P6, and P7 with their three most similar antimicrobial peptides (AMPs) from the ADP database was presented in Table 4. Notably, P7 differed from the known AMP AP02806 by only one amino acid residue, whereas P1 and P6 exhibited relatively lower similarities of 42.31% to 46.67%. The 3D structures of these peptides were predicted using the PEP-FOLD3.5 web server (Figure 4), and the best models for the peptides P1 (4fasD), P6 (2w4lE), and P7 (3g8bB) were recognized with the lowest optimized potential for efficient structure prediction (sOPEP) energy (−17.2362, −22.1334, −21.7099). The structures of P1 and P6 were primarily composed of α-helices, as determined by helical wheel projections generated using the Heliquest web server. P1 and P6 displayed a hydrophobic face characterized by at least six contiguous hydrophobic amino acids arranged in a 3–11 helix configuration. In contrast, P7 was predicted to have a 3–10 helix type with an extended coil–coil region at the N-terminus and did not exhibit a defined hydrophobic face. Collectively, these findings indicated that P1 and P6 were novel AMPs with characteristic α-helical structures.

### 2.6. Scanning Electron Microscopic (SEM) Observations

As P6 showed the strongest bactericidal effect in our study, it was selected to be incubated with *E. coli*, followed by observation under SEM to examine the possible changes in the bacterial surface structure caused by P6. The cell wall of the PBS-treated *E. coli* was complete, and the cells were short and full, with clear edges and a smooth surface (Figure 5A,B). After 2 h incubation with P6, the *E. coli* cell surface became adorned with granular protrusions, a phenomenon clearly illustrated in Figure 5C. Subsequently, these cells demonstrated evident signs of membrane rupture, accompanied by the leakage of intracellular materials (Figure 5D). Obviously, P6 significantly disrupted the membrane integrity of the bacteria, which may be the important reason for cell death (Figure 5D).

## 3. Discussion

In recent years, transcriptomic analyses have been increasingly applied to explore a diversity of AMPs from various organisms, such as bacteria [37], plants [38,39], insects [40,41,42,43], copepods [44], crabs [45], fish [8,46], and mollusks [47], but scarce information is obtained from microalgae. In fact, microalgae are generally considered as a good source of natural antibacterial substances, such as terpenoids and polysaccharides [3], while reports about the amino acid-based bactericidal substances are relatively limited. In recent years, amino acid-based bactericidal substances have been occasionally discovered in various algal species through enzymatic hydrolysis methods [48]. Consequently, it remains uncertain whether valuable antimicrobial peptides are widely present in microalgae. For our current goal of exploring AMPs from microalgae, we employ a new strategy to establish a database to predict the homologous AMPs from our de novo assembly of the *A. anophagefferens* transcriptome. Notably, one short peptide selected from the database is found to be a new AMP with strong antibacterial effects. To our knowledge, this is the first informative study on AMPs from microalgae based on large-scale transcriptomic exploration, and it is also demonstrated that the transcriptomic analysis technique is an effective way for discovering the new AMPs of microalgae.

In our preliminary study, we applied the traditional bioinformatics method to identify the potential AMPs, that is, to annotate all the genes based on the existing genome and transcriptome data and then search for possible AMPs, but this strategy was not successful. According to the annotation results, there were no homologs with high scores similar to those known and common AMPs, such as cathelicidin or defensin [49]. We attempted to clone and recombine some possible target genes, which hit AMPs by BLAST analysis but with relatively low scores, while these genes either could not be successfully cloned probably due to the differences between algae strains or that their high GC content increased the difficulty of cloning and protein expression, resulting in no protein product obtained for functional verification. The failure of this strategy in some aspects explains the reason for the limited reports on the identification of algae AMPs in previous studies.

In our opinion, algae *A. anophagefferens* have many bacteria-like properties, such as an extraordinarily high cell abundance and utilization efficiency of DOM, which enable them to compete with bacteria and other phytoplankton species during the formation of brown tides and are likely to produce AMPs to inhibit the growth of other microorganisms as they grow in large numbers. It has been reported that *A. anophagefferens* can produce various metabolites during growth [5]; therefore, it is wondered whether the strategy to find AMPs from the perspective of lytic proteins would be more appropriate. So, algae *A. anophagefferens* were cultured under different conditions to activate the gene expression as much as possible, and then the sequencing data were de novo assembled using high-quality reads to obtain uniqueness. Using Trinity, it was possible to combine all reads from various samples, resulting in a merged assembly that included all transcripts present in the designated sample set, capturing AMP-related gene expression under different conditions [50]. Subsequently, a bioinformatic analysis pipeline combined with different models and screening conditions was applied to explore the candidate AMPs. This high-throughput screening pipeline for AMPs has been developed for some animal species, such as *Conus betulinus* [51], *Heteromastus filiformis* [52], and zebrafish [53], and this was the first time it was used for microalgae AMP screening. As a result, a total of 471 putative AMP sequences with values higher than 0.5 in the CTDD and PAAC models were obtained, the quantity of which was comparable to that which we had previously determined in *C. betulinus* (466 putative AMPs or AMP-derived genes) [51]. These putative peptides range in length from 12 to 121 amino acids, with net charges from 1 to 25, isoelectric points (PI) from 8.02 to 13.14, an amphiphilicity index from 0.14 to 2.55, and normalized hydrophobicity values from −1.82 to 1.34. Their identity with known AMPs ranges from 0% to 93.33%, which indicates a high diversity of the candidate AMPs from microalgae *A. anophagefferens*.

To further exemplify this pipeline, we focused on the C- or N-terminal AMPs and identified a peptide named P7, which was 15 amino acids in length and shared a high similarity with a well-known histone-derived AMP, H4-(86–100) [36]. Unlike other histone-derived peptides, such as H2A and H2B, which are mainly degraded from the N-terminus by pepsin, H4-(86–100) is a C-terminal histone H4-related peptide, which is first identified in rats [36,54], later proven to be a major component of the antimicrobial action of human sebocytes [55]. Similarly, P7 was identified as a C-terminal fragment from a histone H4 homolog (Table 4) and exhibited antibacterial potency against *E. coli* and *P. pastoris*. The similarity in their antimicrobial potencies could be explained by the fact that their only sequential difference was Gln8 in P7 versus Asn8 in H4-(86–100). Therefore, the identification of P7 as a C-terminal histone-derived AMP well exemplified the effectiveness of the new strategy aimed at short peptides possibly integrated inside proteins. This confirmed that our bioinformatics analysis pipeline was feasible for AMP screening in microalgae.

Although the three identified AMPs with antibacterial activity (P1, P6, P7) all exhibit α-helical structures, P7 differs significantly in structure from P1 and P6, suggesting that their mechanisms of antibacterial activity may also differ greatly. As shown in Figure 4, P1 and P6 mainly form a 3–11 helix-type spiral structure, while P7 forms a non-typical α-helix, specifically a 3–10 helix-type spiral followed by an obvious coil tail. H4-(86–100), the homolog of P7, kills bacteria primarily by regulating ATP-dependent DNA rotase and inhibiting DNA rotatase-mediated DNA supercoiling [36]. Since P7 shares a high similarity of its sequence and structure with H4-(86–100), it is believed that their biological activity and mechanism of action should be similar. For the other two peptides, P1 shows the highest similarity to AP03574 [31], which originates from *Scylla paramamosain*. AP03574 contains an α-helix and exhibits significant antibacterial activity against *S. aureus*, *Pseudomonas aeruginosa*, and *E. coli* with MICs of 1.5–3 μM, 1.5–3 μM, and 3–6 μM, respectively. P6 is most similar to AP00723 [34] and AP00447 [35,56], both of which come from the venom of the solitary eumenine wasp. AP00723, known as Decoralin, is a linear cationic α-helix peptide composed of 11 amino acids. It has a broad spectrum of antimicrobial activity with MICs of 40 μM for *S. aureus*, 80–160 μM for *E. coli*, and 40 μM for *Candida albicans*. The other homolog of P6, Anoplin (AP00447), consisting of 10 amino acids residues, has a highly homologous linear α-helix structure and exhibits broad-spectrum antibacterial activity. Its inhibitory activity is stronger against Gram-positive bacteria than against Gram-negative bacteria, with MICs of 5 μg/mL for *S. aureus* and 50 μg/mL for *E. coli*. As illustrated, the hit peptides of P1 and P6 are all short peptides with helical structures, and the two short peptides share only 42.31% to 46.67% similarity to known AMPs (Table 4), which means P1 and P6 are novel α-helix AMPs that only share key amino acids at crucial positions to preserve their structural integrity. Unlike P7, P1 and P6 form only a typical α-helix, and this structure can easily form pores in the cell membrane through the “barrel-stave model” [57] or the “toroidal pore model” [58]. For example, Melittin assumes an α-helical conformation associated with lipid bilayers; after the peptides aggregate, one or two peptides begin to embed more deeply into the membrane. When the deepest embedded peptide is connected to another interface, it forms a pore to the interior of the cell [59]. SEM observations provided direct, intuitive, and clear images of the surface ultrastructure of P6-treated bacteria, showing that a mass of *E. coli* gathered together, with bacterial surfaces overgrown with granules and forming holes, leading to content leakage after treatment with P6 for 2 h (Figure 5). This proved that P6 probaly caused membrane damage directly in a relatively short period of time.

In this study, the ability of P1, P6, and P7 to inhibit different types of bacteria may vary, while their antifungal abilities are more pronounced. It is worth noting that the antifungal activities of P1 and P6 are notably stronger than that of P7, which reinforces the notion that the antibacterial mechanisms of P1 and P6 are significantly different from those of P7. Given the surface structures of bacteria and fungi are significantly different, we currently cannot confirm whether P1 and P6 can kill fungi through the same membrane-disrupting mechanism. It is reported that some α-helix peptides are nonmembrane disruptive peptides, which may pass through the membrane and interact with variable intracellular macromolecular targets through similar mechanisms as traditional antibiotics. For example, PsD1 from peas can interact with the cell cycle control protein cyclin F from *Neurospora crassa* cells to stop the cell cycle [60]. At least, the results demonstrate that microalgal-derived AMPs may have diverse antimicrobial mechanisms, thereby possessing a broad-spectrum antibacterial or antifungal activity. This would be a significant reason why *A. anophagefferens* can inhibit a variety of other microorganisms, aiding in their rapid growth and proliferation to form brown tides.

As a promising antimicrobial peptide, it is essential for a candidate to possess a relatively low MIC, good safety, excellent stability, and broad-spectrum antibacterial capabilities. The development and application of AMPs as novel anti-infection therapeutic agents are rapidly advancing. Several peptides, including LL-37 [61], Melittin [62], Cecropin P1 [63], magainin II [64], indolicidin [65], and ranalexin [66], along with their derivatives, are currently undergoing clinical studies. LL-37, for instance, is a cationic peptide with an α-helical structure produced by human cells. It exhibits a wide range of antimicrobial activities, with MICs of 9– 44.9 µg/mL against *E. coli* [61], 14–128 µg/mL against *S. aureus* [61], and greater than 250 μg/mL against *Candida albicans* [67]. This MIC range is comparable to P6, which has the strongest antibacterial activity in this study, with MICs of 31.25 μg/mL against *E. coli*, 125 μg/mL against *S. aureus* and *M. luteus*, and 62.5 μg/mL against *P. pastoris*. Safety and stability are also crucial factors to consider in the development of AMPs. For example, Melittin is a highly active antimicrobial peptide, while its direct application in clinical therapy is limited due to its high toxicity [68]. In contrast, P6 has been experimentally verified to be non-hemolytic at very high concentrations, indicating its safety profile. In terms of stability, P6 maintained its effectiveness for at least 30 days at 4 °C, with no significant changes in its inhibitory activity against *E. coli* (unpublished data). Consequently, P6 is identified as a broad-spectrum AMP exhibiting potent activity, high safety, and excellent stability. These characteristics position P6 as a promising candidate for diverse applications in the future. Further research is warranted to explore its efficacy in various clinical settings and to optimize its formulation for practical use.

By comparing the activity and screening parameters of P6, a highly effective AMP, with those of six other synthetic peptides, this study identified several key findings that may optimize the screening process for high-activity AMPs. In this study, seven short peptides derived from either the N-terminus or C-terminus of proteins are synthesized. Except for P4, which has two α-helices connected by random crimping, the remaining six peptides have a single α-helical structure (Appendix A). By comparing their physicochemical properties and antibacterial activities, we can identify critical factors determining the antimicrobial efficacy of α-helical AMPs. Among the seven peptides tested, P6 exhibited the strongest antimicrobial activity, followed by P1 and P7. The remaining four peptides show no or only weak activity. The parameters that show significant differences and are most correlated with antibacterial activity are the normalized hydrophobic moment and the amphiphilicity index (Table 2). In detail, P6 has the highest values for both parameters. P7 shares the same amphiphilicity index as P6, but has a normalized hydrophobic moment similar to P1, which is less than half that of P6. When the normalized hydrophobic moment is as low as 0.36, P3 has no antibacterial effect despite having an amphiphilicity index comparable to P1 and P7. Similarly, when the amphiphilicity index is as low as 0.65, P2 does not exhibit antibacterial activity despite having a normalized hydrophobic moment of one. A particularly interesting case is P5, which has parameters similar to P1 but demonstrates only very weak antifungal and antibacterial activity. Therefore, for α-helical AMPs, higher values for both parameters, ideally exceeding one, may indicate greater reliability and higher antibacterial activity. Another point of reference is the amino acid composition at the interface of the α-helical structure. For example, as shown in Figure 4, the spiral wheel of P6 indicates that its hydrophobic amino acids are on the same side, giving it greater amphiphilicity than P1. This is likely the key reason why P6 has stronger antibacterial activity, despite the similar structures of P1 and P6. These observations provide important information for further optimizing high-throughput screening models of microalgae AMPs, especially for α-helical structure peptides with high bactericidal activity.

## 4. Materials and Methods

### 4.1. Algae Strain

The targeted brown tide causative species, *A. anophagefferens*, was isolated from the coastal waters of Qinhuangdao in the Bohai Sea in 2015. The strain was cultured in the appropriate medium formulated with autoclaved seawater filtered through 0.45 μm membranes and maintained at 20 °C under a photon flux density of 75 μmol·m^−2^·s^−1^ and a 12:12 h light–dark cycle. To prepare the samples of transcriptomic analysis, the algae were designed to be cultured under five different conditions, including the nitrogen-limited group (N group), phosphorus-limited group (P group), glucose-added group (C group), light-limited group (L group), and temperature-limited group (T group), and the algal abundance was measured daily by C6 flow cytometry (DB, NJ, USA). For the N, P, and C group, *A. anophagefferens* with a final cell abundance of approximately 5 × 10^6^ cells/L was cultured in modified L1 culture media with 80 μM nitrate–N and 5 μM phosphate–P. After about a week of cultivation, the original nutrients were exhausted, and then an additional 10 μM phosphate–P, 160 μM nitrate–N, and 106 μM glucose–C (final concentration) were, respectively, added into N, P, and C groups. Furthermore, the C group stopped lighting. For the L and T groups, *A. anophagefferens* with a final cell abundance of approximately 5 × 10^6^ cells/L was cultured in modified L1 culture media with 160 μM nitrate–N and 10 μM phosphate–P. After 72 h of culture, the light intensity and temperature were reduced to 5 μmol·m^−2^·s^−1^ and 10 °C in the L and T groups, respectively. After 24 h treatments, *A. anophagefferens* cells from the five groups were collected for RNA extraction. About 50 mL culture medium (about 10^8^ cells) of each group was centrifuged at 6000× *g* for 10 min to collect the sample, and the pellet was prepared for RNA extraction using Trizol according to the manufacturer’s instructions.

### 4.2. Illumina Library Preparation, Sequencing, and Assembly

The quality of total RNA was checked using an Agilent 2100 Bioanalyzer (Agilent, Santa Clara, CA, USA), and the RNA Integrity Number (RIN) of the extracted RNA was over 8.5. The library preparation and sequencing of the algae under different culture conditions were carried out by Novogene Inc. (Beijing, China) using an Illumina HiSeq 2000 platform.

### 4.3. AMPs Prediction

A total of 3425 putative AMP sequences were obtained from the public AMPs Database (APD3, https://aps.unmc.edu/) by the end of December 2022 [4]. Potential AMP sequences were predicted from the protein sequences by employing the Blast program [69] (e-value at 1 × 10^−5^). The alignment hits with less than 80% of the length of the matched AMPs were filtered out. The remaining sequences were then fed into a series of AMP prediction software, namely AMPml V1.2.7 (2021, China, 2021SR0424886; https://github.com/flystar233/AMPml, accessed on 19 November 2024), CAMP [70], and DBAASP [71]. The overlaps with values higher than 0.5 in the CTDD and PAAC models were considered as potential AMPs, and a total of 471 sequences was obtained. As a preliminary study, we put emphasis on the potential AMPs specifically located at the N-terminus or C-terminus, so 52 candidates were further identified. Among them, the only two C-terminus peptides and five selective N-terminus peptides with the shortest lengths were chosen for synthesizing. They were further classified based on annotation information in the APD3 database [4], and their physicochemical traits were examined using Expasy’s ProtParam [72]. Three-dimensional structures were modeled by PEP-FOLD3 (https://mobyle.rpbs.univ-paris-diderot.fr/cgi-bin/portal.py#forms::PEP-FOLD3, accessed on 19 November 2024) [73] and I-TASSER (https://zhanggroup.org/I-TASSER/, accessed on 19 November 2024) [74], and the helical wheel was predicted by Heliquest (https://heliquest.ipmc.cnrs.fr/index.html, accessed on 19 November 2024) [75].

### 4.4. Peptide Synthesis

The peptides selected based on the various prediction tools mentioned above were synthesized using solid-phase peptide synthesis methods at Sangon (Sangon, Shanghai). Then, each peptide was purified to >90% by high-performance liquid chromatography, and the purity was confirmed by mass spectrometry analysis. The peptides were dissolved in acidified distilled water and stored at −80 °C until used in subsequent experiments.

### 4.5. Antibacterial Assays

Antibacterial activities of the synthesized AMPs against *E. coli*, *S. aureus*, *M. luteus,* and *P. pastoris* were examined as previously reported [76] with some modifications. Firstly, microorganisms in the logarithmic phase were centrifuged, washed by PBS (phosphate-buffered saline) for three times, and then resuspended in PBS (about 10^4^ cfu/mL). Subsequently, 50 µL of dissolved peptides were mixed with 50 µL of bacterial suspensions precatively for an incubation at 37 °C for 2 h. PBS was used as the blank control. Finally, a 20 µL mixture was dispensed into a 96-well plate containing 200 µL of LB medium for *E. coli* and *M. luteus*, and *S. aureus* or YPD medium for *P. pastoris*. There were three parallel wells for each peptide. The plates were then placed in a microplate reader (Biotek, Winooski, VT, USA) and incubated at 37 °C. OD600 values were measured every 0.5 h after a shake for ten seconds for a total of 20 h (bacteria) or 48 h (fungus) to generate the growth curves.

All the data were presented as mean ± standard deviation (n = 3) and analyzed with Statistical Package for Social Sciences (SPSS, USA) 26.0 and GraphPad Prism version 9.0.0 for Windows (GraphPad, USA). The significant differences among groups were tested by paired two-tailed Student’s *t*-test and multiple comparisons.

### 4.6. Hemolytic Activity Assay

The peptides were co-incubated with rabbit red blood cells to verify its hemolysis as reference [31]. First, 20 mL of rabbit red blood cells (source leaf organisms, China) were washed several times until the upper phase was clarified and adjusted to a concentration of 5 × 10^9^ cells/mL. Different concentrations of the active short peptide were then added and incubated at 37 °C for 1 h. Physiological saline and DMSO treatments were used as negative and positive controls, respectively. The absorbance at 540 nm was recorded using a microplate reader (Biotek, Winooski, VT, USA), and the hemolysis rate was calculated as follows:(1)$$Hemolysis\;rate\;\%=\frac{\left[{Test\;sample\;}_{OD540}-{Negative\;control\;}_{OD540}\right]}{\left[{control\;}_{OD540}-{Negative\;control\;}_{OD540}\right]}\times 100\%$$

### 4.7. Scanning Electron Microscope (SEM) Observation

The bacteria were pretreated as described in reference [31]. In detail, 500 μL of P6 with a concentration of 2.5 mg/mL was incubated with the bacteria *E. coli* (a final concentration of 5 × 10^7^ CFU/mL). After incubation for a period of 2 h, they were fixed with precooled 5% glutaraldehyde at 4 °C for 2 h, dehydrated in a series of concentrations of ethanol in a critical point dryer (EM CPD300, Wetzlar, Germany), coated with gold, and finally observed under a scanning electron microscope (Zeiss SUPRA 55, Oberkochen, Germany).

## 5. Conclusions

In recent years, intensive studies have been undertaken for searching for natural antimicrobial substances, while limited AMPs information has been obtained from microalgae. Here, we performed de novo transcriptome sequencing of *A. anophagefferens*, a distinct strain isolated from Qinhuangdao, and obtained a list of 471 potential AMPs, among which the only two C-terminus peptides and five N-terminus putative peptides with the shortest lengths were selected and tested in vitro for their antimicrobial activity toward different types of bacteria and fungi; one novel α-helix AMP was found, a good candidate for further development as effective antibiotic therapeutics. Collectively, we have shown the effectiveness of a combination of in silico and experimental approaches to identify novel AMPs in microalgae and demonstrated that the microalgae transcriptome is an underexplored resource of novel antimicrobial substances, that is, our work will be a good resource for both the transcriptome data and AMP discovery in microalgae.

## Figures and Tables

**Figure 1 ijms-25-13736-f001:**
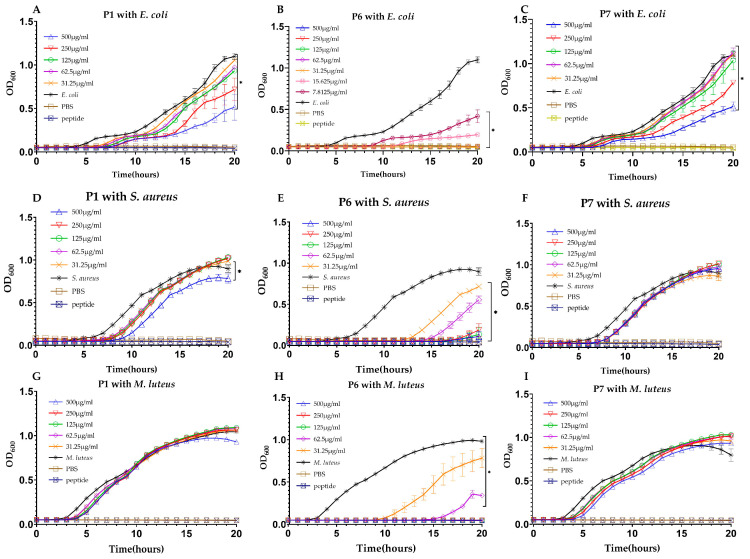
Growth inhibition of various microorganisms by synthesized AMPs. (**A**) *E. coli* growth inhibition by P1. (**B**) *E. coli* growth inhibition by P6. (**C**) *E. coli* growth inhibition by P7. (**D**) *S. aureus* growth inhibition by P1. (**E**) *S. aureus* growth inhibition by P6. (**F**) *S. aureus* growth inhibition by P7. (**G**) *M. luteus* growth inhibition by P1. (**H**) *M. luteus* growth inhibition by P6. (**I**) *M. luteus* growth inhibition by P7. Three control groups were used in the assay. “*E. coli*, *S. aureus, M. luteus*” represents only microorganisms, “PBS” represents only PBS, and each peptide name represents only a single peptide. * denotes significant differences (*p* < 0.01) between the treatment groups and the control group.

**Figure 2 ijms-25-13736-f002:**
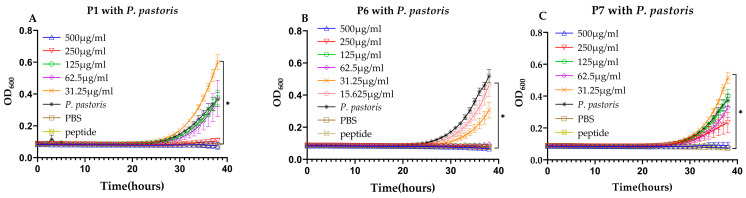
Growth inhibition of *P. pastoris* by synthesized AMPs. (**A**) *P. pastoris* growth inhibition by P1. (**B**) *P. pastoris* growth inhibition by P6. (**C**) *P. pastoris* growth inhibition by P7. Three control groups were used in the assay. “*P. pastoris*” represents only *P. pastoris*, “PBS” represents only PBS, and each peptide name represents only a single peptide. * denotes significant differences (*p* < 0.01) between the treatment groups and the control group.

**Figure 3 ijms-25-13736-f003:**
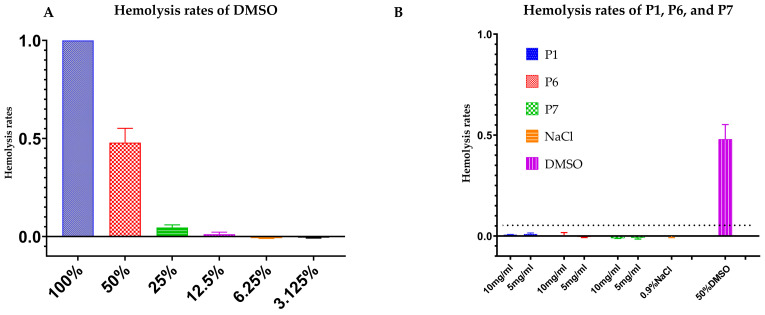
The hemolytic activity of synthesized AMPs. (**A**) Hemolytic activity of DMSO at different concentrations. (**B**) “P1” represents the hemolytic activity of peptide P1, “P6” represents the hemolytic activity of peptide P6, and “P7” represents the hemolytic activity of peptide P7. Two control groups were used in the assay. DMSO treatment was the positive control and 0.9% NaCl treatment was the negative control.

**Figure 4 ijms-25-13736-f004:**
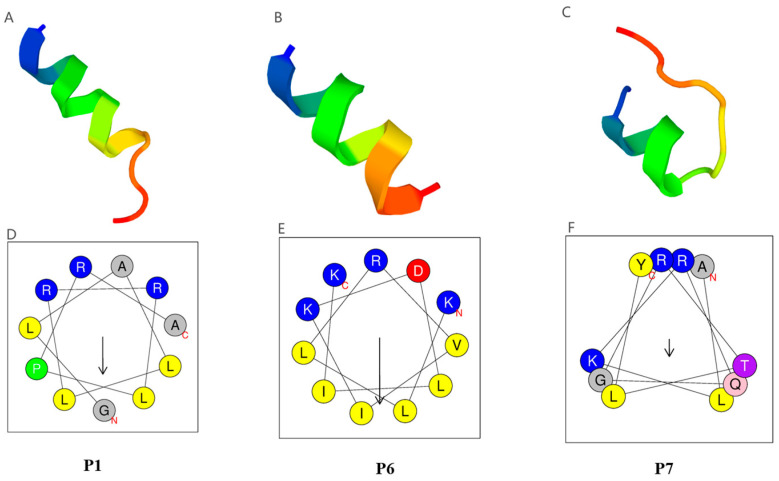
Structure comparison of three peptides. (**A**) The predicted 3D structures of P1 (4fasD as template). (**B**) The predicted 3D structures of P6 (2w4lE as template). (**C**) The predicted 3D structures of P7 (3g8bB as template). (**D**) Helical wheel projections of P1. (**E**) Helical wheel projections of P6. (**F**) Helical wheel projections of P7, where the yellow circles refer to the hydrophobic amino acid residues, the blue ones to the cationic charged residues, the purple circles to the polar uncharged residues, the red circle to an aspartate residue, the pink circle to a glutamine residue, and the green circle to a proline residue. The black arrows indicate hydrophobic faces of the amphipathic structures.

**Figure 5 ijms-25-13736-f005:**
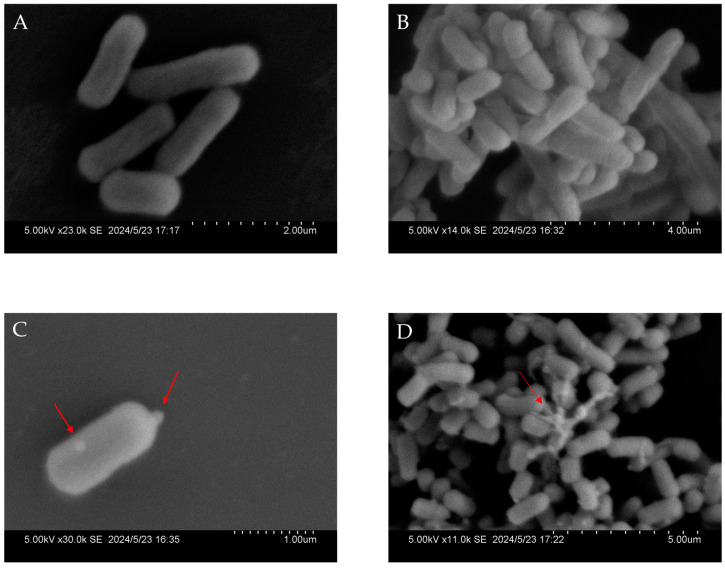
SEM observations of P6-treated *E. coli*. (**A**,**B**) The effect of PBS on *E. coli* observed at 2 h. (**C**,**D**) The effect of P6 on *E. coli* observed at 2 h. The red arrows indicate granular protrusions and the leakage of intracellular materials.

**Table 1 ijms-25-13736-t001:** Raw data statistics of Illumina sequencing.

Sample	Raw Reads	Clean Reads	Clean Bases	Error Rate	Q20	Q30	GC Content
N_1	42,652,334	41,591,096	6.24 G	0.03	95.71	89.58	66.9
P_1	43,142,014	42,374,664	6.36 G	0.03	95.93	90.02	67.18
C_1	41,218,308	40,457,850	6.07 G	0.03	95.10	87.98	66.82
L_1	41,090,120	40,347,984	6.05 G	0.03	95.69	89.48	66.49
T_1	38,559,818	37,772,058	5.67 G	0.03	95.87	89.85	66.13

N_1: nitrogen-limited group (N group); P_1: phosphorus-limited group (P group); C_1: glucose-added group (C group); L_1: light-limited group (L group); T_1: temperature-limited group (T group). Q30, base call accuracy = 99.9%; Q20, base call accuracy = 99%.

**Table 2 ijms-25-13736-t002:** The physicochemical and structural traits of the selected seven peptides.

AMPs	P1	P2	P3	P4	P5	P6	P7
Length	16	17	17	18	18	12	15
Net Charge	5	4	3	5	5	4	3
Isoelectric Point	12.7	12.57	12.1	12.28	12.42	11.49	10.62
Normalized Hydrophobic Moment	0.88	1.00	0.36	0.19	0.85	1.95	0.85
Linear Moment	0.26	1.00	0.34	0.2	0.32	0.31	0.56
Amphiphilicity Index	0.77	0.65	0.81	1.18	0.89	1.33	1.33
Propensity to PPII Coil	1.1	1.03	0.92	1.08	0.98	0.93	0.91
AMP Length	16	17	17	18	18	12	15

**Table 3 ijms-25-13736-t003:** Three synthesized AMPs exhibited antimicrobial activity.

AMPs	Sequence	Antibacterial Activity		Antifungal Activity	
		*E. coli*	*S. aureus*	*M. luteus*	*P. pastoris*
P1	RGLALLRRLPRASPRS	*	*	N	*
P6	RKLLRVIKDLIK	*	*	*	*
P7	VVYALKRQGRTLYGF	*	N	N	*

* represents that this peptide exhibited apparent antimicrobial activity; “N” stands for weak or no antimicrobial activity. Gram-negative bacterium: *E. coli*; Gram-positive bacteria: *S. aureus* and *M. luteus*; yeast: *P. pastoris*.

**Table 4 ijms-25-13736-t004:** Blast similarities of the three peptides against the APD3 database.

AMPs	Hit1	Hit1Similarity	Hit2	Hit2Similarity	Hit3	Hit3Similarity	Remarks
P1	AP03574 [31]	45.45%	AP02938 [32]	43.75%	AP03098 [33]	42.31%	The sequence is rich in R, L, and P.
P6	AP02141	46.67%	AP00723 [34]	46.15%	AP00447 [35] 3D (2MJQ)	46.15%	The sequence might form a helix.
P7	AP02806 [36]	93.33%	AP02805	86.67%	AP03129	40%	The sequence might form a helix.

## Data Availability

The data presented in this study are available on request from the corresponding author. The data are not publicly available due to follow-up experiments are still ongoing.

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
