# Peer review of "A Potent Antibacterial Peptide (P6) from the De Novo Transcriptome of the Microalga Aureococcus anophagefferens"

_ijms, 2024, doi:10.3390/ijms252413736_

Round 1
Reviewer 1 Report
Comments and Suggestions for Authors
In this article, Zhang et al. present a study that combines in-silico and experimental approaches to predict antimicrobial peptides (AMPs). The authors explore a relatively less investigated area by identifying AMPs from microalgae. Through a multi-stage screening process, they shortlisted seven peptides, one of which (P6) demonstrated strong inhibitory effects against various bacterial strains tested in this study. Overall, this is a commendable and innovative piece of work. However, there are several aspects that could be improved to enhance the clarity and robustness of the study. I recommend publication after addressing the following comments:
1. Expand the acronym "AMP" in the abstract for clarity.
2. Provide a reference for lines 77–78.
3. Add citations for statements in lines 90–94.
4. Cite relevant sources for the information in lines 100–105.
5. Spell out "PBS" the first time it appears in the text.
6. The peptide screening strategy needs more explanation:
o Define what is meant by "low-quality alignment."
o Specify if there was a cutoff for the number of sequences considered.
o Clarify the rationale behind selecting peptides lysated from only N- or C-terminal proteins and why the shortest five peptides with a charge >3 were chosen.
7. Improve the readability of figure legends in Figures 1 and 2 by increasing the font size or resolution.
8. Cite the PEP-FOLD3.5 and Heliquest web servers appropriately.
9. Rephrase the sentence in lines 284–286 for better readability.
10. Apart from the mentioned sentences, please review the whole manuscript to add all the important citations.
Author Response
Comments 1: Expand the acronym "AMP" in the abstract for clarity. |
Response 1: Thank you for pointing this out. The full name of “AMP” has been added in the abstract. |
Comments 2: Provide a reference for lines 77–78. |
Response 2: Thanks for the suggestion. We have examined the MS again, and added the corresponding references when it is needed. Comments 3: Add citations for statements in lines 90–94. Response 3: Thanks for the advice. The part describing the specific steps for obtaining antimicrobial components from Saccharina longicruris by enzymatic hydrolysis, is listed as reference 12 after the previous sentence. As it seems not to be conducive to reading and understanding, the relevant literature is quoted again at the end of the descriptions. Comments 4: Cite relevant sources for the information in lines 100–105. Response 4: Thanks for the advice. In the revised manuscript, three references have been added in the corresponding positions (see lines 103-107). Comments 5: Spell out "PBS" the first time it appears in the text. Response 5: Thanks for the suggestion. The full name "phosphate-buffered saline" has been added to the first acronym "PBS" in the revised manuscript (see lines 189-190). Comments 6: The peptide screening strategy needs more explanation: a) Define what is meant by "low-quality alignment." Response 6a: Thanks for the suggestion. “low-quality alignment” refers to the alignment hits that are less than 80% of alignment lengths, but obviously this description is not accurate, and we will change it to “alignments of less than 80% of alignment lengths” in the revised document (see line 167). b) Specify if there was a cutoff for the number of sequences considered. Response 6b: Thank you for your question. Firstly, AMP sequences were predicted from by employing Blast program with e-value cutoff at 1e-5 and the alignment hits length cutoff at 80%. Then, the overlaps of AMP prediction software namely AMPml, CAMP and DBAASP with values higher than 0.5 in the CTDD and PAAC models were considered as potential AMPs. To make it clearer, we have added the later parameters description in the revised MS. c) Clarify the rationale behind selecting peptides lysated from only N- or C-terminal proteins and why the shortest five peptides with a charge >3 were chosen. Response 6c: Thank you for your question. Firstly, we did not identify any typical full length antimicrobial peptide (AMP) molecules within the transcriptome. Additionally, based on our experience, short peptides that cleave from the N- or C-terminus often exhibit antibacterial activity, such as histone-derived AMPs. As an exploratory study aimed at validating the feasibility of our screening process and to enhance the likelihood of discovering highly active antimicrobial peptides, we focused on short peptides with a certain charge that cleave from the N- or C-terminus. To reduce costs, we selected the seven shortest peptides for synthesis and activity validation. Comments 7: Improve the readability of figure legends in Figures 1 and 2 by increasing the font size or resolution. Response 7: Thanks for the suggestion. The legends for Figures 1 and 2 have been enlarged (see lines 227 and 236). Comments 8: Cite the PEP-FOLD3.5 and Heliquest web servers appropriately. Response 8: Thanks for the suggestion. The URL for PEP-FOLD3.5 has been added to the revised version, https://mobyle.rpbs.univ-paris-diderot.fr/cgi-bin/portal.py#forms::PEP-FOLD3 (see line 494). And cited the Heliquest web server related literature (see line 496). |
Response to Comments on the Quality of English Language |
Point 1: Rephrase the sentence in lines 284–286 for better readability. |
Response 9: Thanks for the suggestion. In the revised version, it has been rewritten as “In recent years, amino acid-based bactericidal substances have been occasionally discovered in various algal species through enzymatic hydrolysis methods [49]. Consequently, it remains uncertain whether valuable antimicrobial peptides are widely present in microalgae.” (see line 293-297). |
Additional clarifications |
Thank you again for your attention and review of my manuscript. Your professional knowledge and careful review have greatly helped me to improve the quality of my paper. I would like to remind that I have added a new table (Table 3) in the revised draft, showing the amino acid sequence and antibacterial activity statistics of P1, P6 and P7. At the same time, the reference list has been updated. |
Reviewer 2 Report
Comments and Suggestions for Authors
Aim, significance and novelty
ü Based on scarcity of research on marine microalgal antimicrobial peptides, the current study used Aureococcus anophagefferens as a model to predict AMPs through high-throughput methods, and identified 471 putative bioactive peptides based on de novo transcriptome technique.
ü Among them, 5 shortest polypeptides lysed from the N-terminus, and 2 C-terminus polypeptides were selected to be chemically synthesized for function verification, producing 3 antibacterial activie (P1, P6, and P7) against Escherichia coli, Staphylococcus aureus, Micrococcus luteus and yeast Pichia pastoris, while showing no hemolytic activity at high concentrations up to 10 mg/mL.
ü One peptide (P6) exhibited the most significant antibacterial activity, with the lowest MIC of 31.25 μg/mL against E. coli, which likely causes cell death by directly destroying the bacterial cell membrane.
1. This study may enrich the database of microalgal AMPs and provides an efficient strategy for searching and validating microalgae source AMPs by combining computer analysis with biological activity experiments.
2. The title needs to be clear and in a more concise form. For example it can be: (Potent Antibacterial Peptide (P6) from the de novo Transcriptome of the Microalga Aureococcus anophagefferens)
3. The sentence L29-31, should be reformulated to a less confirmative form. One suggestion may be (The current study may enrich the database of microalgae AMPs and demonstrate an efficient strategy for searching and validating microalgal AMPs by combining computer analysis with biological activity experiments.)
4. The starting point of this work may depend on the work of Gobler, et al., 2011 who indicated that A. anophagefferens possesses genes that may help it outcompete other phytoplankton. (Gobler, C. J., Berry, D. L., Dyhrman, S. T., Wilhelm, S. W., Salamov, A., Lobanov, A. V., ... & Grigoriev, I. V. (2011). Niche of harmful alga Aureococcus anophagefferens revealed through ecogenomics. Proceedings of the National Academy of Sciences, 108(11), 4352-4357). Please refer to that in the introduction section.
5. The first paragraph of the introduction section is giving an example of COVID-19 which is a virus while the objective of the research is to find effective natural antibacterial substances. So, please remove this incompatibility.
6. The amino acid sequence of the three antibacterial peptides (P1, P6, and P7) should be displayed and commented and related to the measured activities.
7. Brief information of these three peptides (P1, P6, and P7), including the number and sequence of amino acids, charge and isoelectric points should be added in the abstract.
8. The authors should add a brief explanation of the de novo transcriptome technique and how it was used to identify the 471 putative peptides.
9. Results are written in the present tense form. It should be in the past tense, since the present tense means it is a fact and so far we cannot claim that.
10. Figure 2. The legends are too small to be read and the font should be increased.
11. Figure 3. The symbols P1, P6 and P7 should be explained in the figure caption.
12. Being a new technique, give a little information on Illumina sequencing in Table 1, explaining the significance of the traits presented in the table.
13. In Table 2 give information on the sizes of the different peptides.
Minor corrections
L21-23, change into (Among them, the five shortest polypeptides lysed from the N-terminus, and only two C-terminus polypeptides were selected to be chemically synthesized for function verification.)
L23, change (are) to (were)
L24, change (and show) to (and showed)
L25, change (exhibits) to (exhibited)
L27, change (observations show)
L52, change (are thought to be the most promising) to (are the most promising)
Author Response
Comments 1: This study may enrich the database of microalgal AMPs and provides an efficient strategy for searching and validating microalgae source AMPs by combining computer analysis with biological activity experiments. |
Response 1: Many thanks for your comments. |
Comments 2: The title needs to be clear and in a more concise form. For example it can be: (Potent Antibacterial Peptide (P6) from the de novo Transcriptome of the Microalga Aureococcus anophagefferens) |
Response 2: Thanks for the advice. We have changed the title as: “A Potent Antibacterial Peptide (P6) Screened from the de novo Transcriptome of the Microalga Aureococcus anophagefferens. Comments 3: The sentence L29-31, should be reformulated to a less confirmative form. One suggestion may be (The current study may enrich the database of microalgae AMPs and demonstrate an efficient strategy for searching and validating microalgal AMPs by combining computer analysis with biological activity experiments.) Response 3: Thanks for the suggestion. We have revised it as suggestions (see lines 30-32). Comments 4: The starting point of this work may depend on the work of Gobler, et al., 2011 who indicated that A. anophagefferens possesses genes that may help it outcompete other phytoplankton. (Gobler, C. J., Berry, D. L., Dyhrman, S. T., Wilhelm, S. W., Salamov, A., Lobanov, A. V., ... & Grigoriev, I. V. (2011). Niche of harmful alga Aureococcus anophagefferens revealed through ecogenomics. Proceedings of the National Academy of Sciences, 108(11), 4352-4357). Please refer to that in the introduction section. Response 4: Thanks for the advice. We have added the description of the relevant research, and added the literature citations (see lines 132-134). Comments 5: The first paragraph of the introduction section is giving an example of COVID-19 which is a virus while the objective of the research is to find effective natural antibacterial substances. So, please remove this incompatibility. Response 5: Thank you for your correction. We have reorganized this part as the suggestion in line 38-49. Comments 6: The amino acid sequence of the three antibacterial peptides (P1, P6, and P7) should be displayed and commented and related to the measured activities. Response 6: Thank you for your comments. Table 3 is added in the revised manuscript to show the amino acid sequences and activities of P1, P6 and P7. Comments 7: Brief information of these three peptides (P1, P6, and P7), including the number and sequence of amino acids, charge and isoelectric points should be added in the abstract. Response 7: Thank you for your advice. Due to the limited length of the abstract, we try our best to add length and charge number of P6 in abstract, and other detailed information is listed in Table 2, especially adding amino acids number of all peptides to update Table 2. Comments 8: Results are written in the present tense form. It should be in the past tense, since the present tense means it is a fact and so far we cannot claim that. Response 8: Thank you for your corrections. The description of the result has been changed to the past tense (see lines 321-334, 339-341, 345-352, 382-387, 431-432). Comments 9: Figure 2. The legends are too small to be read and the font should be increased. Response 9: Thanks for the suggestion. The legends of Figures 1 and 2 have been revised (see lines 227 and 236). Comments 10: Figure 3. The symbols P1, P6 and P7 should be explained in the figure caption. Response 10: Thanks for the advice. We added "P1" represents the hemolytic activity of peptide P1, "P6" represents the hemolytic activity of peptide P6, "P7" represents the hemolytic activity of peptide P7 (see lines 244-245). Comments 11: Being a new technique, give a little information on Illumina sequencing in Table 1, explaining the significance of the traits presented in the table. Response 11: Thanks for the suggestion. In the revised manuscript, the parameters of Illumina sequencing in Table 1 are explained (see lines 157-162). Comments 12: In Table 2 give information on the sizes of the different peptides. Response 12: Thank you for advice. The length of each peptide has been added in Table 2 (see line 210).
|
Response to Comments on the Quality of English Language |
Point 1: L21-23, change into (Among them, the five shortest polypeptides lysed from the N-terminus, and only two C-terminus polypeptides were selected to be chemically synthesized for function verification.) L23, change (are) to (were) L24, change (and show) to (and showed) L25, change (exhibits) to (exhibited) L52, change (are thought to be the most promising) to (are the most promising) |
Response 13: Thank you very much for your patient correction. We have corrected these errors in the revised MS, and examined the MS again carefully by different person. |
Additional clarifications |
Thank you again for your attention and review of my manuscript. Your professional knowledge and careful review have greatly helped me to improve the quality of my paper. I would like to remind that I have added a new table (Table 3) in the revised draft, showing the amino acid sequence and antibacterial activity statistics of P1, P6 and P7. At the same time, the reference list has been updated. |
Reviewer 3 Report
Comments and Suggestions for Authors
The paper fit with aim and scopes of the International Journal of Molecular Sciences, however I have the following major remarks to the manuscript:
1. Authors use immediately in the Abstract of the manuscript AMPs abbreviation but they even do not introduce this abbreviation in the text of article.
2. In the first sentence of the Abstract section authors said that research on AMPs is limited due to technical limitations? But they do not give any examples, what kind of limitations?
3. The beginning of Introduction section is a mess. Authors speak one time for AMPs and the other for many illnesses without giving any examples.
4. I do not agree with the postulated in line 41 that some herbal drugs are very easily available. Extraction of active substances from herbal is one of the most difficult and study process.
5. I would like to inform authors that there are very informative and full review from Vladkova et al. on the topic of peptides isolated from marine sources which is not cited here:
6. I really could not evaluate the obtained results for antimicrobial activity of the isolated peptides. The figures 1 and 2 are so small that nobody could see anything on these figures. Even zoom of 600% did not give me a chance to see and evaluate the obtained results.
In my opinion this work has scientific impact but it needs of serious revision and improvement of figures quality and/or their separation in order to be possible to be seen and evaluated.
Comments on the Quality of English LanguageEnglish language of the paper is very difficult to be red. There are sentences of 5-6 lines which are absolutely incomprehensible (see lines 45-51 and 52-58). I recommend to authors first of all to use short sentences and secondly some natural English speaker to check the English language of the manuscript.
Author Response
Comments 1: Authors use immediately in the Abstract of the manuscript AMPs abbreviation but they even do not introduce this abbreviation in the text of article. |
Response 1: Thanks for the correction. The full name of “AMP” has been added in the abstract. |
Comments 2: In the first sentence of the Abstract section authors said that research on AMPs is limited due to technical limitations? But they do not give any examples, what kind of limitations? |
Response 2: Thank you for your question. This description in the abstract has been modified (see lines 20-22). Comments 3: The beginning of Introduction section is a mess. Authors speak one time for AMPs and the other for many illnesses without giving any examples. Response 3: Thank you for pointing that out. We have rewritten the beginning part of the Introduction (see lines 38-49). Comments 4: I do not agree with the postulated in line 41 that some herbal drugs are very easily available. Extraction of active substances from herbal is one of the most difficult and study process. Response 4: Thank you for your comments. First of all, we agree that it is difficult to extract active substances from herbs, but in Chinese traditional medicine, there are many herbs that can be directly used. For example, the identification and extraction of artemisinin are complex processes, and the research that led to its discovery was awarded the Nobel Prize. However, sweet wormwood (Artemisia annua L.), the source of artemisinin, is a traditional Chinese herb that is commonly decocted in water for use in reducing fever and treating symptoms of malari. From this perspective, the acquisition and use of herbal remedies can be quite straightforward. Comments 5: I would like to inform authors that there are very informative and full review from Vladkova et al. on the topic of peptides isolated from marine sources which is not cited here: https://journal.uctm.edu/node/j2023-5/JCTM_2023_58_1_22-117_pp825_839.pdf Response 5: Thank you for your advice. We have added relevant descriptions and references to the article in the introduction (see lines 78-79). Comments 6: I really could not evaluate the obtained results for antimicrobial activity of the isolated peptides. The figures 1 and 2 are so small that nobody could see anything on these figures. Even zoom of 600% did not give me a chance to see and evaluate the obtained results. Response 6: Thanks for the comments. The legend for Figures 1 and 2 has been enlarged (see lines 227 and 236). |
Additional clarifications |
Thank you again for your attention and review of my manuscript. Your professional knowledge and careful review have greatly helped me to improve the quality of my paper. I would like to remind that I have added a new table (Table 3) in the revised draft, showing the amino acid sequence and antibacterial activity statistics of P1, P6 and P7. At the same time, the reference list has been updated. |
Round 2
Reviewer 2 Report
Comments and Suggestions for Authors
The revised version is in a good state for publication. The authors responded positively and conveniently to all comments
Reviewer 3 Report
Comments and Suggestions for Authors
Authors improved their manuscript and now it could be accepted for publication.